# Detection of *Salmonella* Pathogenicity Islands and Antimicrobial-Resistant Genes in *Salmonella enterica* Serovars Enteritidis and Typhimurium Isolated from Broiler Chickens

**DOI:** 10.3390/antibiotics13050458

**Published:** 2024-05-16

**Authors:** Tsepo Ramatla, Ntelekwane G. Khasapane, Lungile N. Mlangeni, Prudent Mokgokong, Taole Ramaili, Rendani Ndou, Jane S. Nkhebenyane, Kgaugelo Lekota, Oriel Thekisoe

**Affiliations:** 1Centre for Applied Food Safety and Biotechnology, Department of Life Sciences, Central University of Technology, 1 Park Road, Bloemfontein 9300, South Africa; tramatla@cut.ac.za (T.R.); snkheben@cut.ac.za (J.S.N.); 2Unit for Environmental Sciences and Management, North-West University, Potchefstroom 2531, South Africa; 28844009@mynwu.ac.za (L.N.M.); prudent.mokgokong@nwu.ac.za (P.M.); rendani.ndou@nwu.ac.za (R.N.); lekota.lekota@nwu.ac.za (K.L.); oriel.thekisoe@nwu.ac.za (O.T.); 3Department of Animal Health, School of Agriculture, North-West University, Mmabatho 2735, South Africa; 23892854@nwu.ac.za

**Keywords:** broiler chickens, *S.* Enteritidis and Typhimurium serovars, antimicrobial resistance genes (ARG), pathogenicity islands

## Abstract

Rapid growth in commercial poultry production is one of the major sources of *Salmonella* infections that leads to human salmonellosis. The two main *Salmonella enterica* serovars associated with human salmonellosis are enteritidis and typhimurium. The aim of this study was to determine the prevalence of *S. enterica* serovars Enteritidis and *S.* Typhimurium as well as their *Salmonella* pathogenicity islands (SPI) and antibiotic resistance profiles in broiler chicken feces from slaughterhouses. A total of 480 fecal samples from broiler chickens that were grouped into 96 pooled samples were identified to have *Salmonella* spp. using the *invA* gene, whilst the *Spy* and *sdfI* genes were used to screen for the presence of *S.* Enteritidis and *S.* Typhimurium serovars, respectively, by polymerase chain reaction (PCR) assays. The isolates were also screened for the presence of *Salmonella* pathogenicity islands (SPIs) using PCR. The disc diffusion assay was performed to determine the antibiotic resistance profiles of the isolates. A total of 36 isolates were confirmed as *Salmonella* spp. through amplification of the *invA* gene. Out of 36 confirmed *Salmonella* spp. a total of 22 isolates were classified as *S*. Enteritidis (n = 8) and were *S.* Typhimurium (n = 14) serovars. All (n = 22) *S.* Enteritidis and *S.* Typhimurium isolates possessed the *hilA* (SPI-1), *ssrB* (SPI-2) and *pagC* (SPI-11) pathogenicity islands genes. Amongst these serovars, 50% of the isolates (n = 11/22) were resistant to tetracycline and nalidixic acid. Only 22% of the isolates, *S.* Typhimurium (13.6%) and *S.* Enteritidis (9.1%) demonstrated resistance against three or more antibiotic classes. The most detected antibiotic resistance genes were *tet*(*K*), *mcr-1*, *sulI* and *strA* with 13 (59.1%), 9 (40.9%), 9 (40.9%) and 7 (31.8%), respectively. The findings of this study revealed that *S.* Typhimurium is the most prevalent serotype detected in chicken feces. To reduce the risk to human health posed by salmonellosis, a stringent public health and food safety policy is required.

## 1. Introduction

*Salmonella* is a member of the Enterobacteriaceae family, and it is facultatively anaerobic, Gram-negative, oxidase-negative, rod-shaped, mobile, and exhibits peritrichous flagellation [1]. It was named after D.E. Salmon, an American bacteriologist and veterinarian who isolated “hog cholera bacillus” together with T. Smith in 1885 [2]. A variety of gastrointestinal disorders can be caused by *Salmonella* microorganisms, which frequently colonize the intestinal tract of humans and animals [3]. Globally, *Salmonella enterica* Typhimurium and Enteritidis are the most common *Salmonella* serotypes causing gastroenteritis in humans [4]. The serotypes Enteritidis and Typhimurium can be isolated from chickens before slaughter and from humans who have become ill after eating infected chicken meat [5]. Globally, poultry production is expanding rapidly to meet demand [6]. A chicken contaminated with *Salmonella* spp. is considered unfit for human consumption [7]. Salmonellosis outbreaks are frequently linked to chicken products because chickens are carriers of the *Salmonella* bacterium in their guts with potential for vertical transmission [8].

The virulence factors result in bacteria invading, adhering, and replicating inside host cells. Virulence factors also enable bacteria to cause disease by overcoming the host defences [9]. *Salmonella* has several known virulence factors, which are found in *Salmonella* pathogenicity islands (SPIs), prophages, fimbrial clusters and plasmids [10]. Most of the virulence genes involved in pathogenesis are located in the SPI-1 and SPI-2 pathogenicity islands in the genome of *Salmonella* [11]. Due to the requirement of SPI-1 and the type II secretion system (T3SS) for pathogenicity, *Salmonella* strains lacking SPI-1 and SPI-2 are unable to generate intestinal inflammation [12,13].

While pathogenicity islands enable bacteria to induce disease, the resistance islands are genomic islands that offer antimicrobial resistance to antibiotics which still promotes the advantage of bacterial infection to cause disease. Genetically encoded antibiotic resistance can be considered a subtype of virulence factors as they promote host pathogenesis, thereby allowing chronic diseases [14,15].

The global rise of antibiotic resistant (AR) bacteria is a major public health problem [1]. Antibiotic resistance genes (ARGs) have emerged in human and animal infections due to the excessive use of antimicrobial drugs in animals raised for food, frequently without the advice or supervision of a specialist [16,17]. The first report of *Salmonella* resistance to a single antibiotic was published in the early 1960s [18]. One of the causes for the emergence of AR might be due to the use of antimicrobials for metaphylaxis, prophylaxis, treatments, and growth promotion [19]. *Salmonella* Typhimurium and non-Typhimurium isolates have been reported to develop antibiotic resistance, especially in resource-poor countries [20,21]. Several studies have been conducted in South Africa on the prevalence of *Salmonella* in chickens [6,20,21,22]. A study done in the North West Province, South Africa, discovered that chicken samples acquired from various retail shops were contaminated with *Salmonella* [6]. However, there is a scarcity of data regarding the prevalence, virulence, and antibiotic resistance of *Salmonella* of broiler chickens at slaughterhouses in South Africa. Therefore, the objective of this study was to determine the prevalence of *S.* Enteritidis and *S.* Typhimurium serovars, SPIs, and antibiotic resistance from faecal samples of broiler chickens collected from the slaughterhouse in the North West Province of South Africa.

## 2. Results

### 2.1. Prevalence of S. Typhimurium and S. Enteritidis Serovars

A total of 96 pooled fecal samples of broiler chickens were used for isolation of *S.* Typhimurium and *S.* Enteritidis serovars using XLD agar, resulting in a total of 48 non-repetitive presumptive isolates. Amplification of the invA *Salmonella* genus-specific gene revealed that 75% (36/48) of the isolates were *Salmonella* spp. All the 36 Salmonella isolates were further screened for the presence of *S.* Typhimurium and *S.* Enteritidis serovars targeting *Spy* and *sdfI* genes, respectively. Eight (22.2%) isolates were confirmed as *S.* Enteritidis, while fourteen (38.9%;) were *S.* Typhimurium (Figure 1). The *16S rRNA* gene sequence analysis of the *S.* Typhimurium and *S.* Enteritidis revealed a high percentage of nucleotide similarity (99.9%) to the reference NCBI GenBank sequences of the respective. The representative isolates were deposited in GenBank under accession numbers OR416208 and OR416209 for *S.* Typhimurium as well as (OR416957 and OR41695) for *S.* Enteritidis.

### 2.2. Detection of Virulence Genes in S. Typhimurium and S. Enteritidis

Twenty-two *S.* Typhimurium and *S.* Enteritidis isolates, that were positively confirmed by PCR, were further screened for the presence of the following virulence genes: *hilA*, *ssrB*, *marT*, *sopB*, *pagN*, *vexA*, *nlpI*, *bapA*, *pagC*, *oafA*, *spvB* and *cdtB*. None of the isolates harbored the *pagN* gene of the SPI-1 (Figure 2). All the 22 isolates (*S*. Typhimurium and *S.* Enteritidis) possessed *hilA* (SPI-1), *ssrB* (SPI-2) and *pagC* (SPI-11) genes. The *bapA* (SPI-9), *sopB* (SPI-5), *marT* (SPI-3), *vexA* (SPI-7), *nlpI* (SPI-7), and *oafA* (SPI-12) genes were present in 8 (36.4%), 7 (31.8%), 5 (22.7%), 4 (18.2%), 4 (18.2%) and 3 (13.6%) of the two *Salmonella* serovars, respectively. While the chromosomal *cdtB* and plasmid *spvB* genes were detected from 6 (27.3%) and 4 (18.2%) isolates, respectively.

### 2.3. Phenotypic Detection of Antibiotic Resistance Profiles

Antimicrobial susceptibility testing of the 22 isolates demonstrated a wide range of antimicrobial resistance profiles against only the nine drugs tested. Fifty percent of the isolates were resistant to tetracycline and nalidixic acid. This was followed by eight (36.4%), six (27.3%), six (27.3%), five (22.7%), four (18.2%), four (18.2%), and two (9.1%), isolates that were resistant to colistin sulphate, ciprofloxacin, chloramphenicol, streptomycin, ampicillin, amoxicillin-clavulanic acid, and gentamicin, respectively. All the isolates were fully susceptible to the cefepime antibiotic (Figure 3). Five isolates, that is, *S.* Typhimurium n = 3 (13.6%) and *S.* Enteritidis n = 2 (9.1%) were multidrug-resistant (MDR) as they demonstrated resistance to three or more antibiotic classes. All intermediate isolates were considered to be susceptible.

### 2.4. Genotypic Detection of Antibiotic Resistance Profiles

The data presented in Figure 4 shows the successful amplification of the antibiotic resistance genes in *S.* Typhimurium and *S.* Enteritidis isolates. A total of 22 isolates possessed *tet*(*K*), *tet*(*O*), *tet*(*A*) and *tet*(*K*), 13 (59.1%), 6 (27.3%), 4 (18.2%) and 3 (14.5%), respectively; these genes are associated with tetracycline resistance. Additionally, 9 (40.9%) and 8 (36.2%) of the isolates harbored colistin resistance genes *mcr-1* and *mcr-4*, respectively. Furthermore, some isolates possessed aminoglycoside resistance associated genes *strA* n *=* 7 (31.8%), *strB* n = 4 (18.2%), *aadA* n = 4 (18.2%) and *aadE* n = 1 (4.5%), while others harbored sulphonamides resistance genes; *sulI* (40.9%) and *sulII* (4.5%), respectively. For the beta-lactamase, 32 isolates possessed *CTX-M* (45.5%), *ampC* (36.4%), *TEM* (36.4%), *OXA* (18.2%) and *SHV* (9.1%), respectively. Antibiotic resistance genes for tetracycline (*tet*(*X*), *tet*(*P*)), colistin (*mcr-2*, *mcr-3*, *mcr-5*), sulphonamides (*sulIII*) and *CARB* were not amplified in all screened isolates.

### 2.5. Coexistence of Phenotypic and Genotypic Antibiotic Resistance Traits

Out of eleven isolates that showed phenotypic resistance to tetracycline using phenotypic disc diffusion assay, n = 2 isolates carried *tet*(*A*), n = 4 *tet*(O), n = 8 *tet*(*W*) and n = 1 *tet*(*K*) genes encoding tetracycline resistance. Moreover, three isolates that showed phenotypic resistance to colistin harbored the *mcr-1* gene, while n = 4 isolates carried the *mcr-4* gene. Three isolates with phenotypic resistance to amoxicillin carried *TEM*, n = 1 harbored both *OXA* and *CTX-M* genes. Three isolates showed phenotypic resistance to streptomycin and carried the *strA* gene, while one isolate harbored *strB* gene. One isolate with positive phenotypic resistance to ampicillin carried both *CTX-M* and *ampC* genes.

## 3. Discussion

*Salmonella* Enteritidis and *S.* Typhimurium serovars are frequent causes of foodborne illness and death. The *Salmonella* pathogenicity island 1 (SPI-1) and the type 3 secretion system (TTSS) contain a potential inner membrane component called the invasion protein A gene (*invA*) [23]. It affects the host cell by delivering type III secreted effectors and is required for the invasion of epithelial cells [23]. The *invA* gene has been used in numerous studies to identify and confirm *Salmonella* spp. [8,24,25,26,27]. Since it has been established that the *invA* gene is only found in *Salmonella* species, molecular diagnostic techniques can be used to successfully confirm the existence of this genus [28,29]. In order to ascertain the prevalence of *Salmonella* species in broiler chickens, the *invA* gene was employed in this study. As a result, 75% (36/48) of the isolates were identified as *Salmonella*. This finding is higher than the previous prevalence reported in Türkiye and Egypt, whereby 38.2% and 20% of *Salmonella* serovars were isolated from chickens, respectively [30,31].

The *Salmonella* isolates (36/48) in this study were further screened for the presence of *S.* Enteritidis and *S.* Typhimurium, where eight (22.2%) isolates were confirmed as *S.* Enteritidis, while 14 (38.9%) were *S*. Typhimurium. A comparative study conducted in Egypt reported a low prevalence of *S.* Typhimurium (3%) and *S.* Enteritidis (2%) serovars in chicken meat [32]. Another study conducted in Egypt recorded a low prevalence of *S.* Enteritidis (9%), while the same study recorded high prevalence of *S.* Typhimurium (86.6%) in chickens [23]. These are further supported by observations in poultry meat, where only 9% (36/400) of the isolates were positive for *S.* Typhimurium and *S.* Enteritidis [3]. However, the only limitation of this study is that selective enrichment broths such as Rappaport–Vassiliadis broth or Muller–Kauffmann tetrathionate used to increase the isolation rate of *Salmonella* was not used.

The present study also sought to determine the occurrence of SPIs from *S.* Typhimurium and *S.* Enteritidis isolated from poultry meat. The SPI-1 (*hilA*) gene was detected in all isolates in this study. The invasion of epithelial cells by *Salmonella* is promoted by SPI-1 [25]. In *Salmonella* infections, these effectors play a variety of roles, including rearranging the host cytoskeleton, recruiting immune cells, regulating cell metabolism, secreting fluid, and regulating inflammation [33]. The *hilA* gene directly induces the expression of two SPI-1 genes (*invF* and *sicA*) that encode SPI-1 T3SS apparatus components. The SPI-1 T3SS effectors that are encoded both inside and outside SPI-1 are activated by *invF*, a transcriptional activator of the *AraC* family [25,34]. All the isolates in this study possessed the SPI-2 (*ssrB*) and SPI-11 (*pagC*) pathogenicity islands. The SPI-2 secretes effectors to promote formation of an intracellular *Salmonella*-containing vacuole (SCV), which provides *S.* Typhimurium with an ideal environment to replicate [35]. The *ssrB* gene is autoregulated, and it activates SPI-2 and SPI-2 co-regulated genes in response to unknown signals [36,37]. The genes encoded on SPI-13 play an important role in intracellular viability. A cluster in SPI-13 encodes putative lyase, hydrolase, oxidase, and arylsulphatase regulators, and deletion of this island attenuates *Salmonella*'s ability to reproduce [12,38]. Furthermore, the SPIs such as SPI-9 (*bapA*), SPI-5 (*sopB*), SPI-3 (*marT*), SPI-7 (*vexA*), nlpI (SPI-7), and SPI-12 (*oafA*) were detected in few isolates in this study. This indicates that all isolates of both *S.* Enteritidis and *S*. Typhimurium serovars are very important in human and animal infections.

The cytolethal distending toxin B (*cdtB*) was detected in 27.3% in this study. This gene has previously been detected in 4.92% of isolates obtained from children with salmonellosis in China [22]. In the cytolethal distending toxin (CDT) complex, *cdtB* is the active form, with a similar amino acid sequence to that of DNase I in mammals [39,40]. The protein homologue of the CDT active component is encoded by the gene *cdtB*. The common bacterial infection that produces this toxin damages DNA, causing cell cycle arrest and cellular distension [41]. The *spvB* gene that encodes for an enzyme that ADP-ribosylates actin and destabilizes the cytoskeleton of eukaryotic cells was also detected at a low percentage. Three virulence-related genes are found in the *spv* region, including the transcriptional regulator *spvR* and the two structural genes *spvB* and *spvC*, of which *spvB* is the most significant [42,43,44]. The *spvB* gene is believed to contain two functional domains based on homologies of two separate protein classes [45]. The *spvB* gene has been identified as an ADP-ribosyl transferase that stimulates the breakdown of host cell actin, which in turn causes cytotoxicity in macrophages and pathogenicity in mice [46].

Primarily, antimicrobial drugs are essential to both human and animal health and survival on a global scale [17]. In the Alborz Province of Iran, Sodagari et al. [47] first introduced tetracycline as the most efficient antibiotic against *Salmonella* in chickens. The results of the present study showed that half of *S.* Typhimurium and *S.* Enteritidis strains showed the greatest level of antibiotic resistance to tetracycline and nalidixic acid. These results are lower than the results obtained in a previous study conducted by Nazari Moghadam et al. [3], where 72.2% and 61.2% of the identified *Salmonella* isolates were resistant to tetracycline and nalidixic acid, respectively. Furthermore, the study conducted in Iran [48] documented 100% resistance to nalidixic acid and 92.3% resistance to tetracycline. The high resistance to the antibiotics contradicts the previous study which reported a low prevalence of resistance against nalidixic acid (14.3%) [49] from chicken samples in the North West province of South Africa. These differences could be explained by geographical location and sample types. Colistin sulphate, ciprofloxacin, chloramphenicol, streptomycin, ampicillin, amoxicillin–clavulanic acid, and gentamicin were detected at low percentages; eight (36.4%), six (27.3%), six (27.3%), five (22.7%), four (18.2%), four (18.2%), and two (9.1%), respectively. This observed prevalence is lower than the reported prevalence from the study conducted by El-Sharkawy et al. [25], whereby 100% of the isolates possessed gentamycin and 89.7% harbored streptomycin. The study conducted by Ezzatpanah et al. [50] in Iran reported the highest rates of nalidixic acid (86.7%) and amoxicillin (45.3%) from poultry samples. This might be because antimicrobial drugs were used more often than usual in the intensive livestock farming system where these studies were conducted.

Moreover, 22% of the isolates demonstrated resistance against three or more antibiotic classes in this study. Studies conducted in Cameroon and Malaysia reported relatively low MDR prevalence of 13% and 23.5% for non-typhoidal *Salmonella* isolates, respectively [51,52]. In contrast, high MDR prevalence of 81.1% and 100% was reported by studies conducted in China and Bangladesh, respectively [53,54].

It was discovered that several drugs evaluated in this study had antibiotic resistance genes associated with them. In this study, four *tet* genes were detected, namely, *tet*(*K*) (59%), *tet*(*O*) (27.3%), *tet*(*A*) (18.2%) and *tet*(*K*) (5.4%). The study by Adesiji et al. [55] in India reported high detection of *tetA* (100%) from *Salmonella* spp. isolated from human, poultry, and seafood sources. In South Africa, more than 70% of antibiotics used in raising livestock can be purchased without a prescription, leading to a rise in antibiotic resistance within the country [19]. Tetracyclines are the most commonly used, or overused, antibiotics in livestock production in South Africa. This is due to the fact that they are extensively accessible and reasonably priced over-the-counter veterinary drugs [56].

Aminoglycoside resistance associated genes *strA*, *strB*, *aadA* and *aadE* were detected in this study at 31.8%, 18.2%, 18.2% and 14.5%, respectively. Similar observations have been reported in previous studies [25,57,58]. The *sul* genes, which code for sulphonamide resistance, were detected in this study. The *sulI* and *sulII* were detected in nine (40.9%) and one (4.5%) of the samples, respectively. This is lower compared to a previous study conducted in Egypt, where 57% of the isolates carried *sul1* gene [25]. The most commonly reported genes among isolates resistant to sulfonamide are *sul1* and *sul2*, which are also present in plasmids of other *Salmonella* species that are still widely distributed in bacterial plasmids that are Gram-negative [19,59]. Furthermore, 40.9% and 36.2% of the *S.* Typhimurium and *S.* Enteritidis isolates possessed *mcr-1* and *mcr-4* genes, encoding for colistin resistance. A study conducted previously in Italy reported that 2.96% *Salmonella* isolates harbored the *mcr-2* gene, while only 1.69% possessed the *mcr-4* gene [60]. Mei et al. [61] reported that only 2.02% of the *Salmonella* strains harbored the *mcr-1* gene in China. This raises serious concern, as antibiotics like colistin and carbapenems are used to treat MDR bacterial infections in humans [62]. None of the isolates used in this study were *tet*(*X*), *tet*(*P*), *mcr-2*, *mcr-3*, *mcr-5*, *sulIII* or *carB* positive. 

Animal and human infections, notably those brought on by *Salmonella* serovars, are frequently treated with beta-lactams [8,63]. During the last decade, *Salmonella* isolates carrying ESBLs have spread worldwide [63]. The beta-lactam-resistant *bla* genes, namely, *CTX-M*, *ampC*, *TEM*, *OXA* and *SHVI* genes are detected in 45.5%, 36.4%, 36.4%, 18.2% and 9.1%, respectively, in this study. Yang et al. [64] detected a higher prevalence of *blaTEM* in 51.6% of resistant *Salmonella* isolates from retail meats at the marketplace in China. The study conducted on retail meat in Canada by Aslam et al. [65], recorded that 17% of the *Salmonella* isolates harbored the *bla_TEM_* gene. In China, 81.2% of the *Salmonella* isolates from chickens carried the *bla_TEM_* gene, although all isolates were negative for the *blaCTX-M* gene [66]. Siddiky et al. [67] detected positive *bla_TEM_* genes in 69.62% *Salmonella* isolates from poultry processing environments in wet markets in Dhaka, Bangladesh. In addition, the study conducted in Central Ethiopia reported the detection of *bla_TEM_*, *bla_TEM-1_*, *bla_TEM-57_*, and *bla_OXA_* in 79% of the animals and human non-typhoidal *Salmonella* isolates [68]. Currently, *bla_CTX-M_* enzymes are the most common type of ESBL because they are derived from the environment [69]. Based on the amino acid composition of the enzyme, the *bla_CTX-M_* enzyme can be divided into five subgroups as follows: *bla_CTX-M-1_*, *bla_CTX-M-2_*, *bla_CTX-M-8_*, *bla_CTX-M-9_*, and *bla_CTX-M-25_* [19]. Globally, the prevalence of carbapenem-resistant Enterobacteriaceae has increased due to the growing use of carbapenems to treat ESBL-producing infections [70].

## 4. Materials and Methods

### 4.1. Sampling

Fecal samples were collected from the ceca/rectum of healthy broiler chickens from four different chicken abattoirs around Mahikeng city of North West Province, South Africa. We randomly collected a total of 480 chicken fecal samples, post-evisceration, from the intestines in slaughterhouses which resulted in 96 pools (5 chickens per pool). Thereafter, the samples were placed in a cooler box and then transported to the laboratory.

### 4.2. Microbiological Analysis

Fecal content (1 g) was weighed and transferred to a sterile container. Then, 10 milliliters of peptone water (BPW Oxoid, Biolab, Johannesburg, South Africa) was added and the mixture was homogenized by vortexing for 2 min and then incubated at 37 °C for 24 h. Thereafter, bacterial cells were streaked onto xylose–lysine–deoxycholate agar (XLD) (Merck, Wadeville, South Africa) after being incubated at 37 °C overnight for 24 h. The colonies were examined for their morphological appearance on the plate (colonies with or without black centers, colorless or opaque-white colonies surrounded by pink or red zones). Three to five colonies were selected per culture and purified on XLD agar and then incubated at 37 °C for 24 h. The Gram staining, catalase, Simmons citrate test, urease, and Triple sugar iron (TSI) agar tests were performed according to Akinola et al. [24]. The Gram-negative rods and the catalase-positive samples were preserved and stored in 20% glycerol (Merck, SA) at −80 °C.

### 4.3. Genomic DNA Extraction

The boiling–centrifugation method was conducted for the extraction of bacterial genomic DNA [71,72]. Briefly, pure colonies of each isolate were aseptically homogenized in 100 μL of sterile distilled water. The suspensions were separately boiled at 100 °C for 15 min and centrifuged at 10,000 rpm for 10 min. Thereafter, the supernatant was transferred to a new microcentrifuge tube and was used as template DNA for polymerase chain reaction (PCR). DNA concentration was measured with a NanoDrop spectrophotometer (ThermoFischer, Waltham, MA, USA). Pure DNA has a 260/280 ratio of between 1.8 and 2.0; phenol and contamination are indicated by a ratio above 2.0, and protein contamination is indicated by a ratio below 1.8.

### 4.4. Molecular Identification of Salmonella Serovars

The PCR assays with following primers were used for amplification of *Salmonella* serovars: for the *invA* gene PCR assay (280 bp) for *S. enterica*; invA-F: GTG AAA TTA TCG CCA CGT TCG GGC AA and invA-R: TCA TCG CAC CGT CAA AGG AAC C [8], *sdfI* gene PCR assay (304 bp) for *S. enterica* serovar Enteritidis using the primers: SdfI -F: TGT GTT TTA TCT GAT GCA AGA GG and SdfI -R: TGA ACT ACG TTC GTT CTT CTG G [73], and *Spy* gene PCR assay (401 bp) for *S. enterica* serovar Typhimurium; Spy-F: TTG TTC ACT TTT TAC CCC TGA A and Spy-R: CCC TGA CAG CCG TTA GAT ATT and also *fliC* gene PCR assay (433 bp) for serovar *S.* Typhimurium fliC-F: CCCCGCTTACAGGTGGACTAC and fliC-R: AGCGGGTTTTCGGTGGTTGT [67]. The reaction volume of 25 μL, contained 12.5 μL PCR Master Mix [AmpliTaq Gold^®^ DNA Polymerase 0.05 units/µL, Gold buffer 930 mM Tris/HCl pH 8.05, 100 mM KCl0, 400 mM of each dNTP and 5 mM MgCl2] (Applied Biosystems, Foster City, CA, USA), 2 μL template DNA, 1 μL of 10 μM each primer utilizing and 8.5 μL nuclease-free water. The thermal cycling was conducted on an Engine T100 Thermal^TM^ cycler (BioRad, Singapore) with the following conditions: an initial step of denaturation at 94 °C for 5 min, then 30 cycles of denaturation at 94 °C for 45 s, annealing at 58 °C for 45 s, and extension at 72 °C for 60 s, followed by a single concluding extension step at 72 °C for 7 min. *Salmonella* Typhimurium (ATCC:14028TM) and *S.* Enteritidis (ATCC:13076TM) were used as positive controls, whilst and *Escherichia coli* (ATCC:259622TM) was used as negative control. PCR products were electrophoresed on a 1.5% (*w*/*v*) agarose gel stained with ethidium bromide and visualized under ultraviolet (UV) light. A 100 bp DNA molecular weight marker (PROMEGA, Madison, WI, USA) was used to determine the size of the PCR amplicons. The Syngene InGenius Bioimager (Cambridge, UK) was used to capture the images.

### 4.5. Identification of Salmonella Serovars Using 16S rRNA

A PCR assay was conducted with bacterial universal primers (27F: AGA GTT TGA TCM TGG CTC AG and 1492R: GGT TAC CTT GTT ACG ACT T) targeting the 16S rRNA gene [74,75]. The PCR conditions were as follows: 96 °C for 4 min for initial denaturation step, followed by 30 cycles of denaturation at 94 °C for 30 s, annealing at 57 °C for 30 s and extension at 72 °C for 1 min, and one step of final extension at 72 °C for 10 min. PCR amplicons were subjected to cycle sequencing using BigDye Terminator cycle sequencing kit (v 3.1) and electrophoresed on the SeqStudio genetic analyzer of the UESM Sequencing facility of North-West University, Potchefstroom. Four representative sequences were submitted to nucleotide Basic Local Alignment Search Tool (BLASTn) (https://blast.ncbi.nlm.nih.gov/Blast.cgi, accessed on 12 October 2023) in order to confirm the isolate’s identity.

### 4.6. Detection of Virulence Genes

To identify the zoonotic potential of the *S.* Enteritidis and *S.* Typhimurium that we have isolated, genomic DNA samples of *Salmonella* isolates were screened for the presence of twelve virulence genes (*hilA*, *ssrB*, *marT*, *sopB*, *pagN*, *vexA*, *nlpI*, *bapA*, *pagC*, *oafA*, *spvB* and *cdtB*) detected by PCR [76]. A PCR mixture of 25 μL, consisted of 8.5 μL nuclease-free water, 12.5 μL 2X PCR Mix (AmpliTaq Gold^®^ DNA Polymerase 0.05 units/µL, Gold buffer 930 mM Tris/HCl pH 8.05, 100 mM KCl0, 400 mM of each dNTP and 5 mM MgCl2) (Applied Biosystems, CA, USA), 2 μL template DNA, and 1 μL of each primer. A DNA free template (nuclease-free water) was included as a negative control. The PCR reactions were subjected to initial denaturation at 94 °C for 5 min, for one cycle, denaturation at 94 °C for 5 min, annealing at (45–66.5 °C) (Table 1) for 45 s; extension at 72 °C for 1 min; and then final extension at 72 °C for 10 min.

### 4.7. Phenotypic Antimicrobial Susceptibility Test

Antimicrobial susceptibility of *Salmonella* isolates was tested for 10 different antimicrobial agents using the Kirby-Bauer disc diffusion method on Mueller Hinton Agar (Oxoid Ltd., Basingstoke, UK). Antibiotics used in this study were 10 μg ampicillin (AMP), 30 μg tetracycline (TET), 10 μg gentamicin (GEN), 5 μg ciprofloxacin (CIP), 25 μg streptomycin (STR), 30 μg cefepime (FEP), 30 μg chloramphenicol (CHL), 30 μg nalidixic acid (NA), 20/10 μg amoxicillin–clavulanic acid (ACA) and 10 μg colistin sulphate (CS). Results were interpreted following the Clinical and Laboratory Standards Institute (CLSI 2018) guidelines coupled with PHE archive based on the European Union protocol for the monitoring of AMR [77]. Resistance to three or more antimicrobials of different classes was considered to be multidrug resistance (MDR) [1].

### 4.8. Genotypic Detection of Antibiotic Resistance Genes

Twenty-four antibiotic resistance genes, tetracycline [*tet*(*A*), *tet*(*O*), *tet*(*X*), *tet*(*P*), *tet*(*W*) and *tet*(*K*)], colistin [*mcr-1*, *mcr-2*, *mcr-3*, *mcr-4* and *mcr-5*], sulphonamides [*sulI*, *sulII* and *sulIII*], aminoglycoside [*strA*, *strB*, *aadA*, and *aadE*] β-lactamase [*ampC*, *SHV*, *OXA*, *CARB*, *TEM* and *CTX-M*], were screened from *S.* typhimurium and *S.* enteritidis isolates using their respective PCR assays (Appendix A). A DNA free template (nuclease-free water) was included as a negative control.

### 4.9. Statistical Analysis

To analyze the data, virulence gene identification, antibiotic susceptibility testing, and antibiotic resistant genes were entered into Excel (Microsoft Excel 2016: Microsoft Corporation, Redmond, DC, USA). The Bubble plot of the antibiotic resistance profile was generated using ChipPlot (https://www.chiplot.online/#, accessed on 8 November 2023).

## 5. Conclusions

The *S.* Typhimurium was the most detected serovar compared to *S.* Enteritidis in fecal samples from broiler chickens. This study has pioneered the detection of SPI genes in *S.* Typhimurium and *S.* Enteritidis isolates found in broiler chicken feces in South Africa. Furthermore, considerable levels of resistance to the colistin and aminoglycosides antibiotic classes were detected in the *S.* Typhimurium and *S.* Enteritidis isolates used in this investigation, raising concerns about a potential risk to human health. Moreover, our findings show that colistin and aminoglycoside resistance is increasing among *Salmonella* species in the region. The recovered *S.* Typhimurium and *S.* Enteritidis isolates possessed MDR and several virulence gene profiles.

## Figures and Tables

**Figure 1 antibiotics-13-00458-f001:**
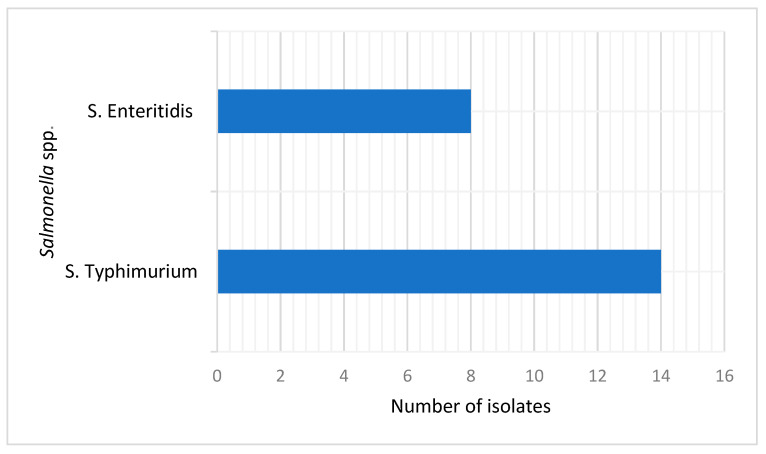
*Salmonella* spp. from pooled fecal samples of broiler chickens.

**Figure 2 antibiotics-13-00458-f002:**
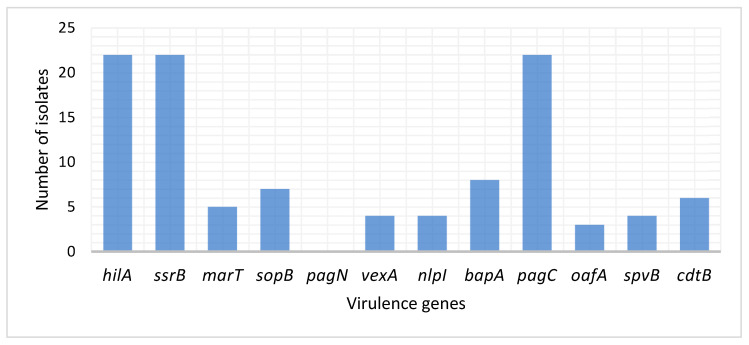
Virulence genes profiles of *S. Typhimurium* and *S. Enteritidis* isolated from the feces of broiler chickens.

**Figure 3 antibiotics-13-00458-f003:**
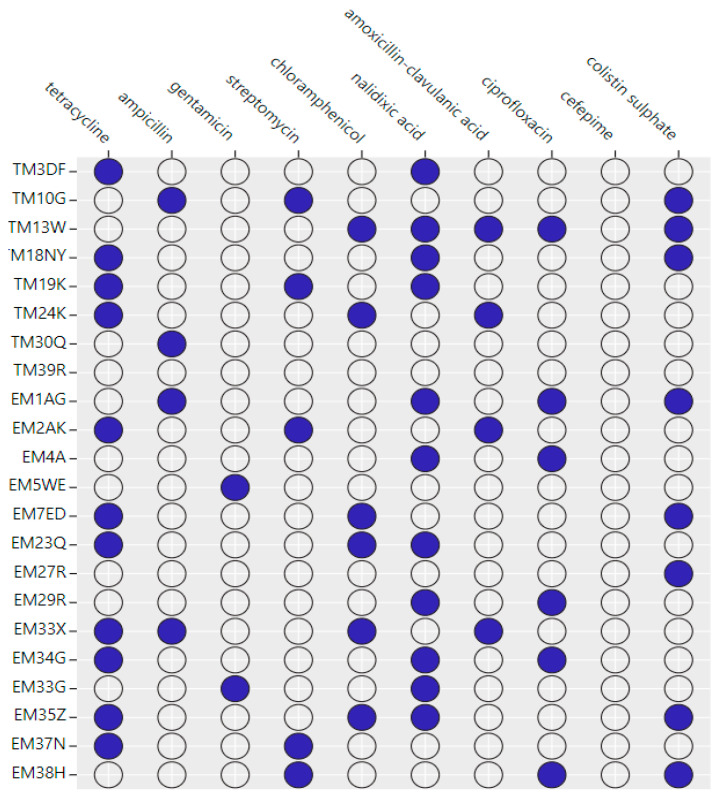
Bubble plot shows the phenotypic antimicrobial resistance profiles detected in 22 isolates of *S.* Typhimurium and *S.* Enteritidis from broiler chickens. The circles shaded blue indicate the resistant antibiotics that were detected in each isolate.

**Figure 4 antibiotics-13-00458-f004:**
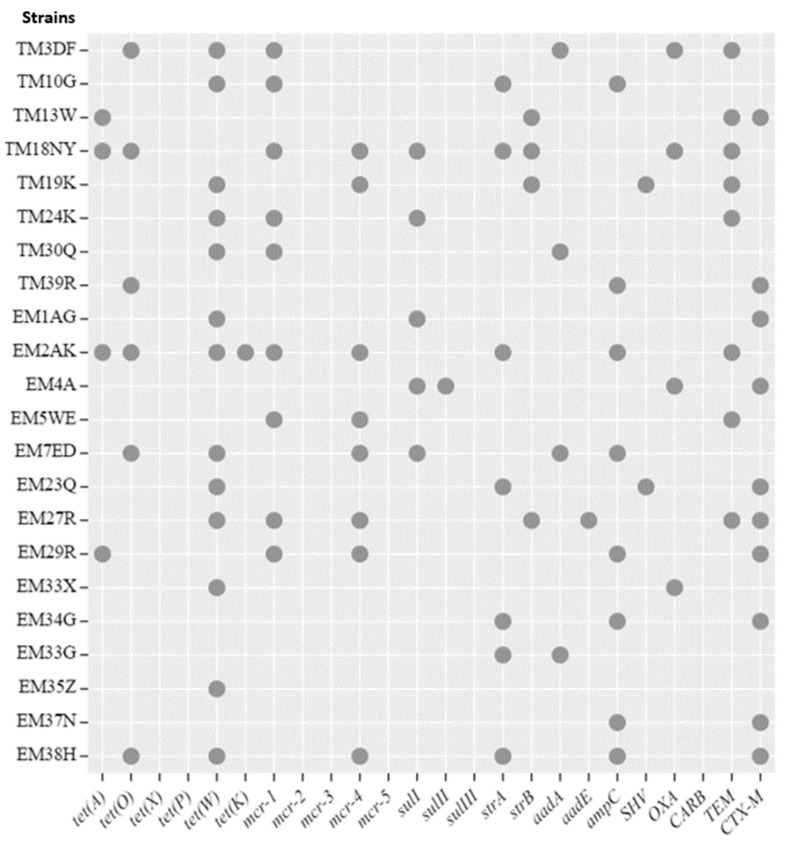
Bubble plot of the antimicrobial resistance gene (ARGs) profiles of *S.* Typhimurium and *S.* Enteritidis isolates from broiler chickens. The grey circles indicate the ARGs that were detected in each isolate.

**Table 1 antibiotics-13-00458-t001:** The oligonucleotide primers used for detection of virulence associated genes of *Salmonella* isolates.

Gene	Location	Primer Name	Primer Sequence (5′-3′)	Amplicon Size (bp)	Annealing Temp (°C)
*hilA*	SPI-1	hilA-F hilA-R	GACAGAGCTGGACCACAATAAGACA GAGCGTAATTCATCGCCTAAAC	312	55 °C
*ssrB*	SPI-2	ssrB-F ssrB-R	CTCATTCTTCGGGCACAGTTA CCTTATTACCCTGGCCTCATTT	558	55 °C
*marT*	SPI-3	marT- F marT-R	CGTCGTCTCACAACAAACATTC CTGACAAATCAATGCCGTAACC	556	55 °C
*sopB*	SPI-5	sopB-F sopB-R	TCACTAAAAACCCAGGAGGCTTTT CGCCATCTTTATTGCGGATTTTTA	1000	65 °C
*pagN*	SPI-6	pagN-F pagN-R	TTCCAGCTTCCAGTACGTTTAG GCCTTTGTGTCTGCATCATAAG	440	55 °C
*vexA*	SPI-7	vexA-F vexA-R	AAACTAAGCGCTCCCGATAC CAGTCGCGCAGTGAAATAATG	504	55 °C
*nlpI*	SPI-8	nlpI-F nlpI-R	AGTCTTGGTTTGAGGGCATTAG TTCTTTCGCCTGCTTCTCATTA	333	55 °C
*bapA*	SPI-9	bapA-F bapA-R	TAAGCGTCGGACTTGGAATG CGTTCTTCAGCGTGTAGGTATAG	543	55 °C
*pagC*	SPI-11	pagC-F pagC-R	CGCCTTTTCCGTGGGGTATGC GAAGCCGTTTATTTTTGTAGAGGAGATGTT	454	66.5 °C
*oafA*	SPI-12	oafA-F oafA-R	CGAGTGACTGGAACCAAAGA CAAGCATAGAGCCAGAGTAGAG	510	55 °C
*spvB*	Plasmid	spvB-F spvB-R	CTATCAGCCCCGCACGGAGAGCAGTTTTTA GGAGGAGGCGGTGGCGGTGGCATCATA	717	66.5 °C
*cdtB*	Genome	cdtB-F cdtB-R	ACAACTGTCGCATCTCGCCCCGTCATT CAATTTGCGTGGGTTCTGTAGGTGCGAGT	268	66.5 °C

## Data Availability

The data and materials of the study will be available from the corresponding author on reasonable request. The sequences of two strains analyzed were deposited in the National Library of Medicine, National Center for Biology Information (NCBI), GenBank nucleotide sequence database. The accession numbers assigned as OR416208 and OR416209 for *S.* typhmurium (https://www.ncbi.nlm.nih.gov/nuccore/ OR416208 and OR416209), OR123649 (https://www.ncbi.nlm.nih.gov/nuccore/ OR123649) and OR416957 and OR41695 for *S.* enteritidis (https://www.ncbi.nlm.nih.gov/nuccore/ OR416957 and OR41695).

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
