# Peer review of "Detection of Salmonella Pathogenicity Islands and Antimicrobial-Resistant Genes in Salmonella enterica Serovars Enteritidis and Typhimurium Isolated from Broiler Chickens"

_antibiotics, 2024, doi:10.3390/antibiotics13050458_

Round 1
Reviewer 1 Report
Comments and Suggestions for Authors
Reviewer Comments
· Abstract – Lines 19-20: Commercial poultry production is growing rapidly globally, which is a source of the Salmonella infections that lead to human salmonellosis.
· Please rephrase as - Rapid growth in the commercial poultry production is one of the major sources of Salmonella infection that leads to human salmonellosis. Otherwise it will give the impression to the readers that it is the only source of human salmonellosis.
· Abstract – Lines 23: … antibiotic resistance profiles in broiler chicken’s faeces … / … antibiotic resistance profiles in broiler chicken faeces …
· Abstract – Lines 24: … faecal samples were identified as Salmonella spp. … / … faecal samples were identified as having/to have Salmonella spp. …
· Abstract – Lines 30-31: Out of 36 confirmed Salmonella spp. a total 30 of 22 isolates, 8 (22.2%) were classified as S. Enteritidis, while fourteen (38.9%) were S. Typhimurium. This sentence is confusing, please rephrase.
· S. Enteritidis and S. Typhimurium understandably are serovars yet you could write them as S. enteritidis and S. typhimurium otherwise if you are writing the serovar after the species name you may feel free to capitalise the first letter. Also in some places it is italicized in other places not.
· Introduction – Line 62: Most of the virulence genes are found in the SPI-1 and SPI-2 found in the genome of … / Most of the virulence genes are found in the SPI-1 and SPI-2 pathogenicity islands in the genome of Salmonella.
· Introduction – Lines 77-79: However, there is a scarcity of data regarding the prevalence, virulence, and antibiotic resistance of Salmonella of at slaughter broiler chickens in South Africa. Please rephrase
· Results 2.3 – Figure 3: The circles shaded blue indicate the present antibiotics that were detected in each isolate. / The circles shaded blue indicate the resistant antibiotics that were detected in each isolate.
· Results 2.3 –Line 130: Genotypic detection antibiotic resistance profiles / Genotypic detection of antibiotic resistance profiles. Also please correct as Results 2.4
· Discussion – Line 157: … Salmonella species in brolier chickens … / … Salmonella species in broiler chickens …
· Discussion: The discussion does not include sufficient information on why there is a prevalence in certain antibiotic resistance genes /resistance to certain antibiotics is observed in this study. Kindly include it to strengthen the discussion.
· Materials and methods – Table 1: Length (bp): Is this the PCR product size?
Comments on the Quality of English LanguageMinor English language editing is required.
Author Response
Abstract – Lines 19-20: Commercial poultry production is growing rapidly globally, which is a source of the Salmonella infections that lead to human salmonellosis.
- Please rephrase as - Rapid growth in the commercial poultry production is one of the major sources of Salmonella infection that leads to human salmonellosis. Otherwise it will give the impression to the readers that it is the only source of human salmonellosis.
Response: Corrected as suggested. Line 19-20, page 1.
- Abstract – Lines 23: … antibiotic resistance profiles in broiler chicken’s faeces … / … antibiotic resistance profiles in broiler chicken faeces …
Response: Corrected as suggested. Line 23, page 1.
- Abstract – Lines 24: … faecal samples were identified as Salmonella spp. … / … faecal samples were identified as having/to have Salmonella spp. …
Response: Corrected as suggested. Line 24, page 1.
- Abstract – Lines 30-31: Out of 36 confirmed Salmonella spp. a total 30 of 22 isolates, 8 (22.2%) were classified as S. Enteritidis, while fourteen (38.9%) were S. Typhimurium. This sentence is confusing, please rephrase.
Response: The sentence was corrected. Line 30-31, page 1.
- S. Enteritidis and S. Typhimurium understandably are serovars yet you could write them as S. enteritidis and S. typhimurium otherwise if you are writing the serovar after the species name you may feel free to capitalise the first letter. Also in some places it is italicized in other places not.
Response: the S. enteritidis and S. typhimurium were corrected throughout the paper.
- Introduction – Line 62: Most of the virulence genes are found in the SPI-1 and SPI-2 found in the genome of … / Most of the virulence genes are found in the SPI-1 and SPI-2 pathogenicity islands in the genome of Salmonella.
Response: The sentence was corrected. Line 62-63, page 2.
- Introduction – Lines 77-79: However, there is a scarcity of data regarding the prevalence, virulence, and antibiotic resistance of Salmonella of at slaughter broiler chickens in South Africa. Please rephrase
Response: Sentence was rephrased as suggested. Line 83-85, page 2.
- Results 2.3 – Figure 3: The circles shaded blue indicate the present antibiotics that were detected in each isolate. / The circles shaded blue indicate the resistant antibiotics that were detected in each isolate.
Response: Corrected as suggested, Line 166-167, page 5.
- Results 2.3 –Line 130: Genotypic detection antibiotic resistance profiles / Genotypic detection of antibiotic resistance profiles. Also please correct as Results 2.4
Response: Corrected as suggested, Line 169, page 5.
- Discussion – Line 157: … Salmonella species in brolier chickens … / … Salmonella species in broiler chickens …
Response: Corrected as suggested, Line 206, page 7.
- Discussion: The discussion does not include sufficient information on why there is a prevalence in certain antibiotic resistance genes /resistance to certain antibiotics is observed in this study. Kindly include it to strengthen the discussion.
Response: The information was added antibiotic resistance. Line276-277, 288-292, page 8, Line 322-327, page 9.
- Materials and methods – Table 1: Length (bp): Is this the PCR product size?
Response: Yes, is PCR product size, we changed Length to “amplicon Size” in Table 1
Reviewer 2 Report
Comments and Suggestions for Authors
CLSI releases documents annually, and there were major changes for Salmonella and Shigella, so why was CLSI 2018 used and not a more recent one? Because several antimicrobials tested do not have reference values for SAlmonella, when tested by disk diffusion (colistin, amoxicillin with clavulanate, gentamicin).
When talking about a Salmonella serovar, it should not be written in italics.
Author Response
Reviewer 2
CLSI releases documents annually, and there were major changes for Salmonella and Shigella, so why was CLSI 2018 used and not a more recent one? Because several antimicrobials tested do not have reference values for SAlmonella, when tested by disk diffusion (colistin, amoxicillin with clavulanate, gentamicin).
Response: It coupled with PHE archive based on the European Union protocol for the monitoring of AMR, and https://doi.org/10.1111/1469-0691.12373
When talking about a Salmonella serovar, it should not be written in italics.
Response: Corrected throughout the paper as suggested.
Reviewer 3 Report
Comments and Suggestions for Authors
This study aimed to determine the prevalence of S.Enteritidis and S.Typhimurium as well as their Salmonella pathogenicity islands (SPI) and antibiotic resistance profiles in broiler chicken’s faeces from Slaughterhouses. However, some areas of concern in this manuscript must be addressed by the authors.
Abstract Line 22: The serovar name is not italicized and starts with a capital letter to distinguish between serovars and species. Check it out across the manuscript. Line 24: the sample size was coming out clearly: It was 480 faecal samples that were divided into 96 pooled broiler faecal samples.
Line 41:Add:antimicrobial-resistant genes (ARG)’’ to the list of keywords Introduction Lines 52-53: Are chickens and humans the only sources of Salmonella? Line 57: Not only for their carrier potential but also for their vertical transmission potential of Salmonella Line 58: Even antibiotic resistance genes are part of the virulence factors Line 62: replace the second ‘‘found’’ with ‘‘located’’ to avoid repetition of the same word Lines 65-66: Make a transitional statement in between the two paragraphs Lines 74-75: Several studies have been conducted in South Africa on the incidence of prevalence of Salmonella in chickens. This statement cannot correct, either you were talking about the incidence or the prevalence of Salmonella but the incidence of the prevalence. There are two different epidemiological concepts. Line 78: correct the following phrase: antibiotic resistance of Salmonella of at… Line 80: correct the following phrase: resistance from faecal samples of broiler chickens collected from the from slaughterhouse Line 87: this is one of the ample examples of cases where Salmonella is italicized across the manuscript. Results Line 139: (). What does this mean Have tried to determine whether there was a perfect matching between the phenotypic and genotypic antibiotic resistance of your isolates? Discussions Lines 159-162: You shouldn’t compare the prevalence of Salmonella in chickens in your study to the prevalence of Salmonella from cattle or milk in previous studies. Lines 167-168: this statement is entirely true since the prevalence of S.Typhimurium (86.65) was instead higher than the one of S.Typhimurium (38.9%) reported in the present study. Line 187: After furthermore, add ‘‘s’’ to ‘‘SPI’’ Lines 215-216: It is not logical to compare the antibiotic resistance profile of Salmonella from two different samples. Methods
Lines 278: Put a full stop after Western Cape. Lines 278-279: Something is missing in the following sentence: province We collected chicken faecal samples randomly post-evisceration from the intestines at slaughterhouses Line 286: Remove ’‘about’’ after ‘‘for’’. Lines 287-288: Why did you not yet transfer the bacterial cells from BPW unto selective enrichment broths such as Rappaport‐Vassiliadis broth, or Muller-Kauffmann tetrathionate to increase the isolation rate of Salmonella given that the isolation rate directs influences the prevalence of Salmonella in the samples. Failure to do this should be highlighted as a limitation of this study in the conclusion section Lines 302-303: How did you assure the purity of the gDNA without using proteinase k and without the washing process (usually with alcohol)?
Lines 304-305: A NanoDrop spectrophotometer was used to measure DNA concentrations. How? Describe briefly the procedure.
Line 317: Never start a sentence with a number: 2 μL template Line 369: Write Sulphonamides in small letters References Check reference 3 for consistency. References 39, 41, 43, 44, 47, 54, 58 are old and need to be replaced with newer ones. For e.g. Matchawe et al (2022)
Comments on the Quality of English Language
There are many grammatical errors, it seems the authors did not proofread the manuscript.
Author Response
Reviewer 3
This study aimed to determine the prevalence of S.Enteritidis and S.Typhimurium as well as their Salmonella pathogenicity islands (SPI) and antibiotic resistance profiles in broiler chicken’s faeces from Slaughterhouses. However, some areas of concern in this manuscript must be addressed by the authors.
Abstract Line 22: The serovar name is not italicized and starts with a capital letter to distinguish between serovars and species. Check it out across the manuscript.
Response: Corrected throughout the paper.
Line 24: the sample size was coming out clearly: It was 480 faecal samples that were divided into 96 pooled broiler faecal samples.
Response: Corrected as suggested, line 24, page 1.
Line 41:Add:antimicrobial-resistant genes (ARG)’’ to the list of keywords
Response: Antimicrobial-resistant genes (ARG) was added, line 41, page 1.
Introduction Lines 52-53: Are chickens and humans the only sources of Salmonella?
Response: Sentence was rephrased. Line 53-54, page 2.
Line 57: Not only for their carrier potential but also for their vertical transmission potential of Salmonella Line 58: Even antibiotic resistance genes are part of the virulence factors
Response: Corrected as suggested, Line 58, page 2.
Line 62: replace the second ‘‘found’’ with ‘‘located’’ to avoid repetition of the same word
Response: found was replace with located as suggested, line 63, page 2.
Lines 65-66: Make a transitional statement in between the two paragraphs
Response: A transitional statement was made. Line 67-71, page 2.
Lines 74-75: Several studies have been conducted in South Africa on the incidence of prevalence of Salmonella in chickens. This statement cannot correct, either you were talking about the incidence or the prevalence of Salmonella but the incidence of the prevalence. There are two different epidemiological concepts.
Response: Corrected. Line 80-80, page 2.
Line 78: correct the following phrase: antibiotic resistance of Salmonella of at…
Response: Sentence was rephrased. Line 84, page 2.
Line 80: correct the following phrase: resistance from faecal samples of broiler chickens collected from the from slaughterhouse
Response: Sentence was rephrased. Line 83-85, page 2.
Line 87: this is one of the ample examples of cases where Salmonella is italicized across the manuscript.
Response: : Corrected throughout the manuscript.
Results Line 139: (). What does this mean Have tried to determine whether there was a perfect matching between the phenotypic and genotypic antibiotic resistance of your isolates?
Response: () was deleted. The perfect matching between the phenotypic and genotypic antibiotic resistance of the isolates is shown in Line 187-195 (2.5), page 6.
Discussions
Lines 159-162: You shouldn’t compare the prevalence of Salmonella in chickens in your study to the prevalence of Salmonella from cattle or milk in previous studies.
Response: corrected. Line 208-210, page 7.
Lines 167-168: this statement is entirely true since the prevalence of S.Typhimurium (86.65) was instead higher than the one of S.Typhimurium (38.9%) reported in the present study.
Response: Corrected, Line 215-217, page 7.
Line 187: After furthermore, add ‘‘s’’ to ‘‘SPI’’
Response: ‘‘s’’ was added, line 239, page 7.
Lines 215-216: It is not logical to compare the antibiotic resistance profile of Salmonella from two different samples.
Response: Corrected. Line 266-269, page 8.
Methods
Lines 278: Put a full stop after Western Cape.
Response: The sentence was deleted.
Lines 278-279: Something is missing in the following sentence: province We collected chicken faecal samples randomly post-evisceration from the intestines at slaughterhouses
Response: Sentence was rephrased. Line 345-347, page 9..
Line 286: Remove ’‘about’’ after ‘‘for’’.
Response: ’‘about’’ was removed as suggested.
Lines 287-288: Why did you not yet transfer the bacterial cells from BPW unto selective enrichment broths such as Rappaport‐Vassiliadis broth, or Muller-Kauffmann tetrathionate to increase the isolation rate of Salmonella given that the isolation rate directs influences the prevalence of Salmonella in the samples. Failure to do this should be highlighted as a limitation of this study in the conclusion section
Response: This limitation was highlighted in Line 219-221, page 7.
Lines 302-303: How did you assure the purity of the gDNA without using proteinase k and without the washing process (usually with alcohol)?
Response: DNA was extracted following the method published by reference 71, 72, and all genomic DNA were successfully amplified including positive control regardless of DNA quality.
Lines 304-305: A NanoDrop spectrophotometer was used to measure DNA concentrations. How? Describe briefly the procedure.
Response: A the procedure NanoDrop was added. Line 367-370, page 10.
Line 317: Never start a sentence with a number: 2 μL template
Response: Corrected. Line 377, page 10.
Line 369: Write Sulphonamides in small letters References Check reference 3 for consistency. References 39, 41, 43, 44, 47, 54, 58 are old and need to be replaced with newer ones. For e.g. Matchawe et al (2022)
Response: Sulphonamides was written in small letter, and new references were added as suggested.
Round 2
Reviewer 2 Report
Comments and Suggestions for Authors
The manuscript is well written and the suggestions were accepted or explained.